

# Diabatic effects on the evolution of storm tracks

Andrea Marcheggiani[1] and Thomas Spengler[1]

[1]Geophysical Institute, University of Bergen, and Bjerknes Centre for Climate Research, Bergen, Norway

**Correspondence:** Andrea Marcheggiani (andrea.marcheggiani@uib.no)

**Abstract.**

Despite the crucial role of moist diabatic processes in midlatitude storm tracks and related model biases, we still lack a more complete theoretical understanding of how diabatic processes affect the evolution of storm tracks. To alleviate this shortcoming, we investigate the role of diabatic processes in the evolution of the Northern Hemispheric storm tracks using a framework based

on the tendency of the slope of isentropic surfaces as a measure of baroclinic development.

We identify opposing behaviours in the near-surface and free troposphere for the relationship between the flattening of the slope of isentropic surfaces and its restoration by diabatic processes. Near the surface (900–800hPa), cold air advection associated with cold air outbreaks initially acts to flatten isentropic surfaces, with air–sea interactions ensuing to restore surface baroclinicity. In the free troposphere (750–350hPa), on the other hand, diabatic generation of slope of isentropic surfaces

precedes its depletion due to tilting by eddies, suggesting the primary importance of moist diabatic processes in triggering subsequent baroclinic development. The observed phasing of the diabatic and tilting tendencies of the slope is observed both in upstream and downstream sectors of the North Atlantic and North Pacific storm tracks, rendering the phasing a general feature of midlatitude storm tracks.

In addition, we find a correspondence between the diabatic generation of slope of isentropic surfaces and enhanced precipita-

tion as well as moisture availability, further underlining the crucial role of moisture and moist processes in the self-maintenance of storm tracks.

## 1 Introduction

Despite the progress in reducing storm track biases over the last decade (Harvey et al., 2020), current climate models still suffer from significant biases in the representation of zonal asymmetries, latitudinal positioning, and overall storm track intensity

(Priestley et al., 2020). Given that diabatic processes significantly influence the evolution of storms and storm tracks (Hoskins and Valdes, 1990; Chang and Orlanski, 1993; Stoelinga, 1996; Swanson and Pierrehumbert, 1997; Chang et al., 2002; Willison et al., 2013; Pfahl et al., 2014; Papritz and Spengler, 2015), it is not surprising that the inadequate representation of moist processes on smaller scales has been identified as a potential key source for these biases (Willison et al., 2013; Pfahl et al., 2014; Schemm, 2023; Fuchs et al., 2023). Yet, we still lack a fundamental understanding of the role of moist diabatic processes

on storm track dynamics. To alleviate this shortcoming, we investigate the impact of diabatic forcing on the evolution of



the baroclinicity in the Northern Hemisphere storm tracks and highlight the crucial role of moist processes for storm track variability.

In the lower troposphere, high surface heat fluxes occur across the strong sea surface temperature (SST) gradients along the western boundary currents (Ogawa and Spengler, 2019). Surface sensible heat fluxes are the primary driver in adjusting surface

air temperatures to the underlying SSTs, thereby establishing and maintaining near-surface baroclinic zones that are argued to anchor storm tracks (Swanson and Pierrehumbert, 1997; Nakamura et al., 2004, 2008; Taguchi et al., 2009; Sampe et al., 2010; Hotta and Nakamura, 2011; Papritz and Spengler, 2015). However, air-sea heat exchange is not always beneficial to storm development, as it can locally dampen temperature contrasts, thereby reducing baroclinicity, particularly in the cold sectors of weather systems (Hoskins and Valdes, 1990; Swanson and Pierrehumbert, 1997; Haualand and Spengler, 2020; Marcheggiani

and Ambaum, 2020; Bui and Spengler, 2021).

In the upper troposphere, diabatic effects associated with latent heat release in cyclones account for the bulk of baroclinicity restoration (Hoskins and Valdes, 1990; Papritz and Spengler, 2015). In fact, cyclones can restore baroclinicity throughout their lifecycle, in particular along their trailing cold fronts where latent heat release due to precipitation leaves behind enhanced baroclinicity for the development of subsequent cyclones leading to cyclone clusters (Weijenborg and Spengler, 2020). It

remains unclear, however, how these diabatic restorations influence storm track activity.

Ambaum and Novak (2014) proposed an idealised model of the storm track, which revealed a predator-prey relationship between meridional heat flux and baroclinicity. This model successfully captures many characteristics of storm track evolution, including the sporadic nature of storm track activity (Messori and Czaja, 2013; Novak et al., 2015, 2017). However, as the model assumes dry dynamics, the restoration of baroclinicity through diabatic processes is represented by a constant external

forcing, without accounting for the direct influence of moist diabatic processes on synoptic time scales. Therefore, the model proposed by Ambaum and Novak (2014) cannot provide insights into the significant role of moist diabatic processes on the spatial structure and positioning of storm tracks (Brayshaw et al., 2009; Papritz and Spengler, 2015). Hence, there remains a need to better understand the role of diabatic processes in the life cycle of storm tracks to aid the development of a more comprehensive model.

We address this gap by clarifying the relationship between adiabatic depletion and diabatic restoration of baroclinicity within the Northern-Hemispheric winter storm tracks by employing the isentropic slope framework (Papritz and Spengler, 2015). The framework distinguishes between adiabatic and diabatic contributions to changes in baroclinicity, which are used to assess their relative importance in the evolution of storm tracks through phase space analysis (Novak et al., 2017; Yano et al., 2020; Marcheggiani and Ambaum, 2020; Marcheggiani et al., 2022). Given the observed near-surface–free troposphere dichotomy

in the maintenance of baroclinicity (Papritz and Spengler, 2015), we partition the troposphere vertically to better highlight the different mechanisms.



## 2 Data

We use the European Centre for Medium-Range Weather Forecasts (ECMWF) ERA-Interim 6-hourly data interpolated onto a $0.5° \times 0.5°$ longitude-latitude grid (Dee et al., 2011). We consider extended winters (November, December, January, February, NDJF) from November 1979 to February 2017. We use instantaneous fields of temperature, geopotential height ($z$), wind velocity (u,v,w), and total column water vapour (TCWV), as well as 6-hourly accumulated precipitation (large-scale and convective) and temperature tendencies due to model physics centred on each time step (as described in Weijenborg and Spengler, 2020). For all fields (except TCWV and precipitation) we use 23 pressure levels (925, 900, 875, 850, 825, 800, 775, 750, 700, 650, 600, 550, 500, 450, 400, 350, 300, 250, 200, 150, and 100 hPa).

We also use the cold air outbreak (CAO) index (as defined in Papritz and Spengler, 2017) and cyclone tracks identified through the University of Melbourne algorithm (Murray and Simmonds, 1991a, b), where we require a minimum duration of at least five 6-hourly steps. For reference, the same tracks were used in Madonna et al. (2020) and Tsopouridis et al. (2021), where additional selection criteria were applied for their analysis.

## 3 Isentropic slope framework for baroclinicity

The framework used in this study was introduced by Papritz and Spengler (2015) and is based on the slope of isentropic surfaces (hereafter referred to as *slope*). The slope $S$ is defined as

$$S \equiv |\nabla_\theta z| \,, \tag{1}$$

where the subscript indicates that the horizontal gradient is taken on an isentropic surface and $z$ is the altitude of the surface. $S$ is a measure of baroclinicity, proportional to the Eady growth rate (Papritz and Spengler, 2015), and can be interpreted as the potential for baroclinic development.

The main advantage of using this framework is the ability to discriminate between different processes changing the slope. The tendency equation of the slope,

$$\frac{DS}{Dt} = \underbrace{\frac{\nabla_\theta z}{S} \cdot \nabla_\theta w_{\mathrm{id}}}_{\text{TILT}} \underbrace{- \frac{\partial z}{\partial \theta} \frac{\nabla_\theta z}{S} \cdot \nabla_\theta \dot{\theta}}_{\text{DIAB}} \underbrace{+ \mathbf{u} \cdot \nabla_\theta S}_{\text{IADV}} \,, \tag{2}$$

features three terms changing the slope. The first term on the right-hand side (TILT) is associated with the tilting of isentropic surfaces by the isentropic displacement vertical wind $w_{id}$ (Hoskins et al., 2003, their Equation 8). The second term (DIAB) describes the deformation of an isentropic surface due to diabatic heating. The TILT and DIAB terms generally exert opposing effects on the slope (Papritz and Spengler, 2015). Finally, the third term (IADV) denotes isentropic advection following the flow, highlighting the fact that slope is generally not a property intrinsic to an air parcel but a characteristic of the surrounding environment. As the IADV term is typically much smaller compared to TILT and DIAB (Papritz and Spengler, 2015; Weijenborg and Spengler, 2020), we neglect it in our analysis.

Isentropic surfaces can become extremely steep, especially closer to the surface within the mixed layer, which leads to numerical issues in the computation of the slope and its tendencies. Therefore, we calculate slope diagnostics only between 900hPa



and 200hPa, additionally masking grid points with low static stability ($\partial\theta/\partial z > 10^{-4}$K m$^{-1}$, as in Papritz and Spengler, 2015). Most of the masking occurs in the lowest layers (900–800hPa), while its effect in the upper troposphere is negligible. The largest amount of masking over the oceans occurs over the western boundary currents (as indicated by stippling in Fig. 1), though it does not affect more than 10-15% of the time steps considered. Masking occurs often over land in correspondence with high orography, though this does not affect our analysis as we exclude land grid points.

To account for the baroclinic structure of midlatitude weather features, it is convenient to separate the vertical averages of the slope and its tendencies into two layers, as different mechanisms are dominant in driving baroclinicity in the lower and upper troposphere. While sensible heating from the ocean restores baroclinicity near the surface (Nakamura et al., 2008; Sampe et al., 2010; Hotta and Nakamura, 2011; Papritz and Spengler, 2015), latent heating associated with cloud and rain formation generates baroclinicity in the free troposphere (Hotta and Nakamura, 2011; Papritz and Spengler, 2015). We thus separate our analysis in the vertical between the near-surface and free troposphere and consider vertical averages across the two partitions separately. Our definition of near-surface troposphere includes pressure levels between 900hPa and 800hPa (every 25hPa), while the free troposphere comprises levels between 750hPa and 350hPa (every 50hPa). The specific partition is made a posteriori and ensues from phase space analysis, which will be discussed in Sect. 4.

Near the surface, the steepest slope (2.5–3 m/km) is found along the strong gradients in SST associated with the oceanic western boundary currents (Fig. 1a). Temperature contrasts along the sea ice edge and low static stability within cold-air outbreaks (CAOs) can also contribute to a steeper slope over the ocean basin (Papritz et al., 2015; Papritz and Spengler, 2017). DIAB is most intense along SST gradients (7–8 m/km day$^{-1}$) and is observed to correspond with strong surface heat fluxes (Fig. 1c). TILT also peaks over the western edge of oceanic basins, especially over areas featuring a high occurrence of CAOs (Fig. 1e).

In the free troposphere, the slope peaks along the main storm track regions in the Northern Hemisphere (2.5–3.25 m/km, Fig. 1b), while DIAB and TILT both feature maximum values along regions of steep slope (Fig. 1d and f, respectively, with values of up to 3 m/km day$^{-1}$ in magnitude), peaking slightly further downstream in both the North Atlantic and North Pacific basins. The slope peaks on the poleward flank of the climatological jet (Fig. 1b), while most intense precipitation and highest storm track density appear to be tightly linked with the strongest DIAB and TILT (Fig. 1d,f).

## 4 Phase space perspective on storm tracks

While the climatology is instructive on the mean structure of midlatitude storm tracks (Fig. 1), it does not provide further insight into the temporal variability associated with the evolution of the two major storm tracks in the Northern Hemisphere. Hence, we consider spatial averages over four regions (see boxes in Fig. 1a) that are expected to behave similarly, both geographically and vertically. Specifically, we look at the upstream sectors, where the slope is higher and the influence of SST gradients is stronger, and downstream sectors, where, despite weaker slope, weather activity is nevertheless intense (Fig. 1f), as storms are generally in a more advanced stage of their lifecycle. The four domains (Fig. 1a, Table 1) represent the upstream (GSE, KOE) and downstream (ENA, ENP) sectors of both the North Atlantic and North Pacific storm tracks.





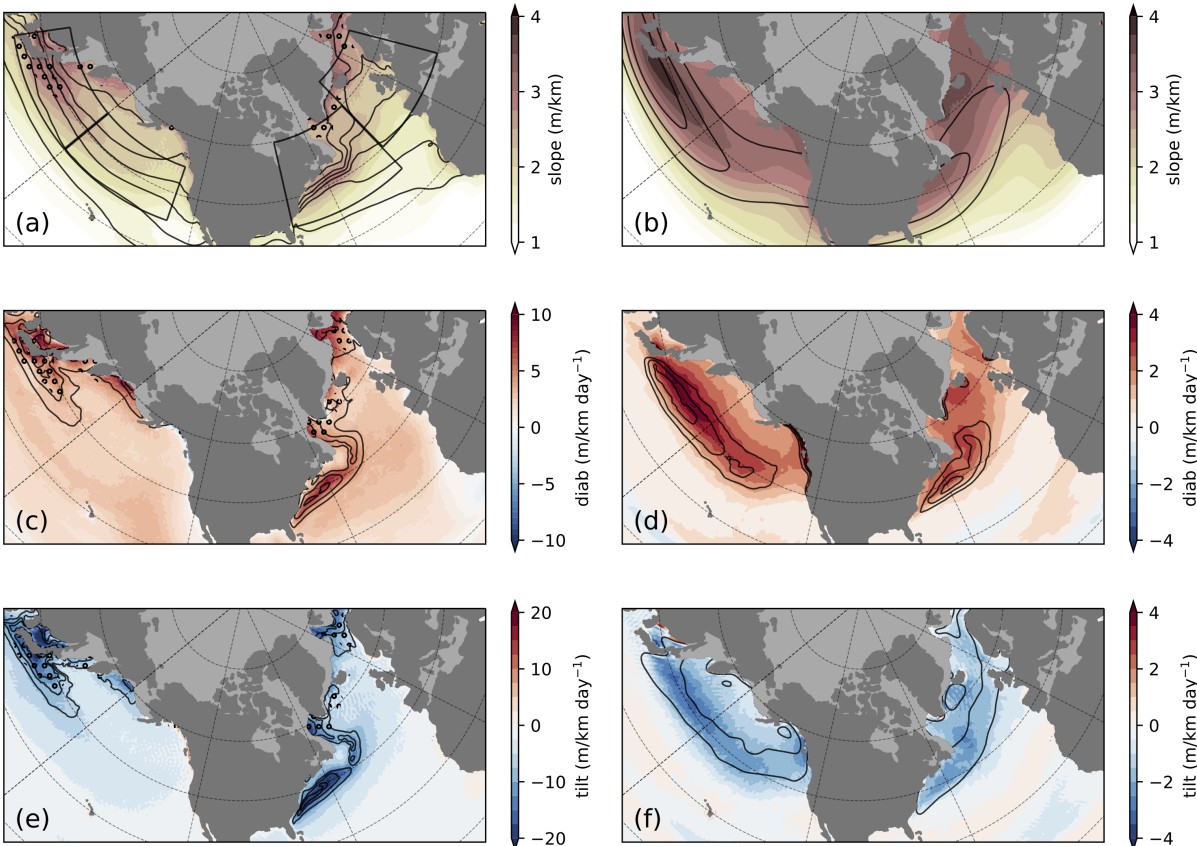

**Figure 1.** Winter (NDJF) climatology of (a,b) isentropic slope $S$, (c,d) DIAB, and (e,f) TILT (shading). Panels (a,c,e) and (b,d,f) show climatological means for the near-surface and free troposphere, respectively. Solid contours represent: (a) SST (every $4°C$ between $4°C$ and $24°C$); (b) wind speed at the dynamical tropopause (i.e., at 2PVU; 30, 40, 50 ms$^{-1}$); (c) surface heat flux (latent + sensible, 50, 75, 100 Wm$^{-2}$, positive upwards); (d) total precipitation (convective plus large-scale; 5, 6, 7 mm day$^{-1}$); (e) CAO index (2, 3, 4 K); (f) cyclone track densities (10, 20, 30 cyclones per season; Gaussian kernel of size 200km used for estimation). Hatching represents areas where slope values are masked for more than 25% of the time (but globally less than 50%). Also marked in (a) are the regional domains for our analysis (see text for details).

## 4.1 Construction of time series

In spatially averaging over the respective domains (Table 1), we exclude land grid points to minimise the effect of orography on the slope diagnostics (Papritz and Spengler, 2015). We also exclude ocean surfaces with sea ice concentration above 15% for more than 5% of the time to limit the effects of inaccuracies in representing sea ice in reanalyses (Renfrew et al., 2021). Finally, we leave out any grid points pertaining to the Mediterranean and Baltic seas to avoid interference from local, mesoscale dynamics. We apply the same exclusion criteria for both the near-surface and free troposphere for the sake of consistency.



**Table 1.** List of the regions used for spatial averaging.

| Acronym | Area | Geographical extent |
|---------|------|---------------------|
| GSE | Gulf Stream Extension | 30º–60º N, 80º–30º W |
| KOE | Kuroshio-Oyashio Extension | 30º–50º N, 130º–180º E |
| ENA | Eastern North Atlantic | 40º–70º N, 30º W–20º E |
| ENP | Eastern North Pacific | 30º–50º N, 180º–130º W |

The time series of the slope, DIAB, and TILT for the 2013-14 winter season are characterised by peaks of intense activity interspersed among periods of weaker-amplitude variability (Fig. 2). This sporadic nature in the temporal variability of the slope and its tendencies is primarily associated with the synoptic evolution that exerts a similar influence on meridional and surface heat fluxes (Messori and Czaja, 2013; Marcheggiani and Ambaum, 2020).

DIAB and TILT generally have opposite signs, both for the sample season as well as for the climatology. There are instances where they change sign, mainly due to orographic effects advected over the ocean (not shown), which are not entirely filtered out by excluding land grid points.

While both DIAB and TILT appear to evolve almost instantaneously, a closer inspection of the time series reveals the existence of a phase shift between the two, hinting at the possibility that one drives. To better understand the observed relationship between DIAB and TILT, we utilise a phase space analysis, which is particularly useful in studying the dynamical evolution of chaotic systems (Novak et al., 2017; Yano et al., 2020; Marcheggiani and Ambaum, 2020; Marcheggiani et al., 2022).

### 4.2 Construction of phase portraits

We construct a phase space, where the x- and y-axis measure the average DIAB and TILT, respectively. We then plot the time series of DIAB against TILT, yielding trajectories in the phase space. Given the length of the time series and its irregular variability, we need to apply a kernel smoother to evince the typical phase space circulation, where no time filtering was applied to the original time series. We use a Gaussian kernel to minimise the amount of noise without losing significant features of the phase space circulation. Our results are qualitatively unchanged for different reasonable choices of the kernel size and refer the reader to Novak et al. (2017) and Marcheggiani et al. (2022) for more details on the applied kernel smoothing.

Once we obtained the average velocity field $\mathbf{u}$ in the phase space, we define a streamfunction $\psi$ to visualise the non-divergent flow $\mathbf{F} = D\mathbf{u}$ (or mass flux, as $D$ represents data density at each point in the phase space), such that

$$\mathbf{F} = (F_x, F_y) = \left(-\frac{d\psi}{dy}, \frac{d\psi}{dx}\right). \tag{3}$$

We thus obtain a phase portrait of the DIAB–TILT co-evolution. If two variables are completely unrelated to each other, the corresponding phase portrait would be extremely noisy and incoherent, regardless of filter strength/size, while for two variables that are directly proportional to each other the circulation would collapse onto a diagonal. Departures from the diagonal are indicative of the existence of a phase shift in their co-evolution, whereby one leads in time. In the extreme case of a 90° phase



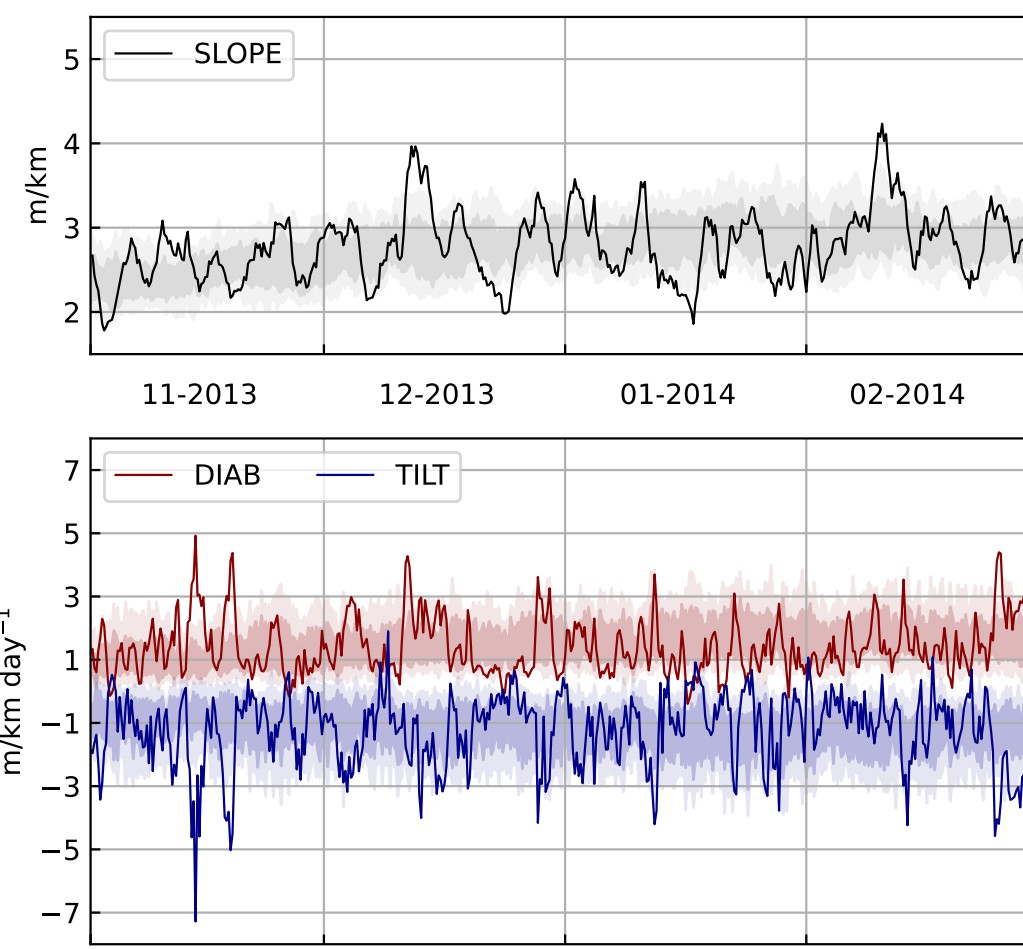

**Figure 2.** Time series of (a) free-tropospheric isentropic slope, and (b) DIAB and TILT, spatially averaged over the GSE region. Solid lines represent a sample season (NDJF 2013-14) while light (dark) shading represents the interdecile (interquartile) ranges over the climatological period (1979–2017).

shift, the resulting phase portrait would feature a perfectly circular circulation. One can also estimate the mean value of any





**Table 2.** Time periods between consecutive stages (i, ii, iii, and iv) as indicated in the phase portraits shown in Fig. 3 for each of the spatial domains (see Table 1), for the near-surface and free troposphere. Last column on the right shows total duration along closed trajectories highlighted in Fig. 3.

| Area | i→ii | ii→iii | iii→iv | iv→i | Tot. duration |
|------|------|--------|--------|------|---------------|
| NEAR-SURFACE TROPOSPHERE (900–825 hPa) | | | | | |
| GSE | 0.79 d | 0.87 d | 1.67 d | 1.90 d | 5.23 d |
| KOE | 1.04 d | 0.74 d | 1.73 d | 1.33 d | 4.84 d |
| ENA | 1.04 d | 1.06 d | 1.40 d | 0.93 d | 4.43 d |
| ENP | 1.54 d | 0.90 d | 1.15 d | 1.49 d | 5.08 d |
| FREE TROPOSPHERE (750–350 hPa) | | | | | |
| GSE | 0.77 d | 0.68 d | 1.09 d | 2.08 d | 4.61 d |
| KOE | 0.97 d | 0.90 d | 0.83 d | 1.67 d | 4.37 d |
| ENA | 0.87 d | 1.25 d | 1.74 d | 0.98 d | 4.83 d |
| ENP | 1.45 d | 1.21 d | 1.54 d | 1.60 d | 5.79 d |

variable across the phase space by calculating its kernel average, where we consider the area-mean slope and calculate its kernel average to inspect its variability in parallel to the evolution of DIAB and TILT (Fig. 3).

155     Inspecting phase portraits for each pressure level individually (not shown), we found a reversal in the phase space mean circulation between 825hPa and 750hPa, transitioning from anticlockwise in the near-surface (Fig. 3a–d) to clockwise in the free troposphere (Fig. 3e–h). We accommodate for this dichotomy by separating our analysis into the near-surface (900-825hPa) and the free troposphere (750–350hPa). Although previous studies already adopted a similar vertical separation (e.g., Papritz and Spengler, 2015, used 600hPa as partition level), our phase space analysis allows for a more dynamically motivated sepa-

160 ration of levels. The circulation changes structure entirely as we reach levels above 350hPa (not shown), where stratospheric dynamics most likely become more relevant and affect the typical circulation.

## 4.3   Phase portraits

For all domains and vertical sections, there is a strong link between DIAB and TILT, as evinced by the near-diagonal alignment of the average circulation. As indicated, the main difference between the near-surface and free-troposphere is the direction of

165 the mean circulation in the phase space, with TILT leading DIAB in the near-surface troposphere and DIAB leading TILT in the free troposphere. Integrating the phase speed along isolines of the streamfunction, we obtain an average duration of one revolution of about 5 days, with slightly shorter cycles in the free troposphere (see Table 2).

    Consistent with the climatology (Fig. 1), the magnitudes of both DIAB and TILT are largest near the surface, in particular over the upstream regions, with average values up to 30 m/km day$^{-1}$ where the SST gradients are strongest (Fig. 3a,b), almost





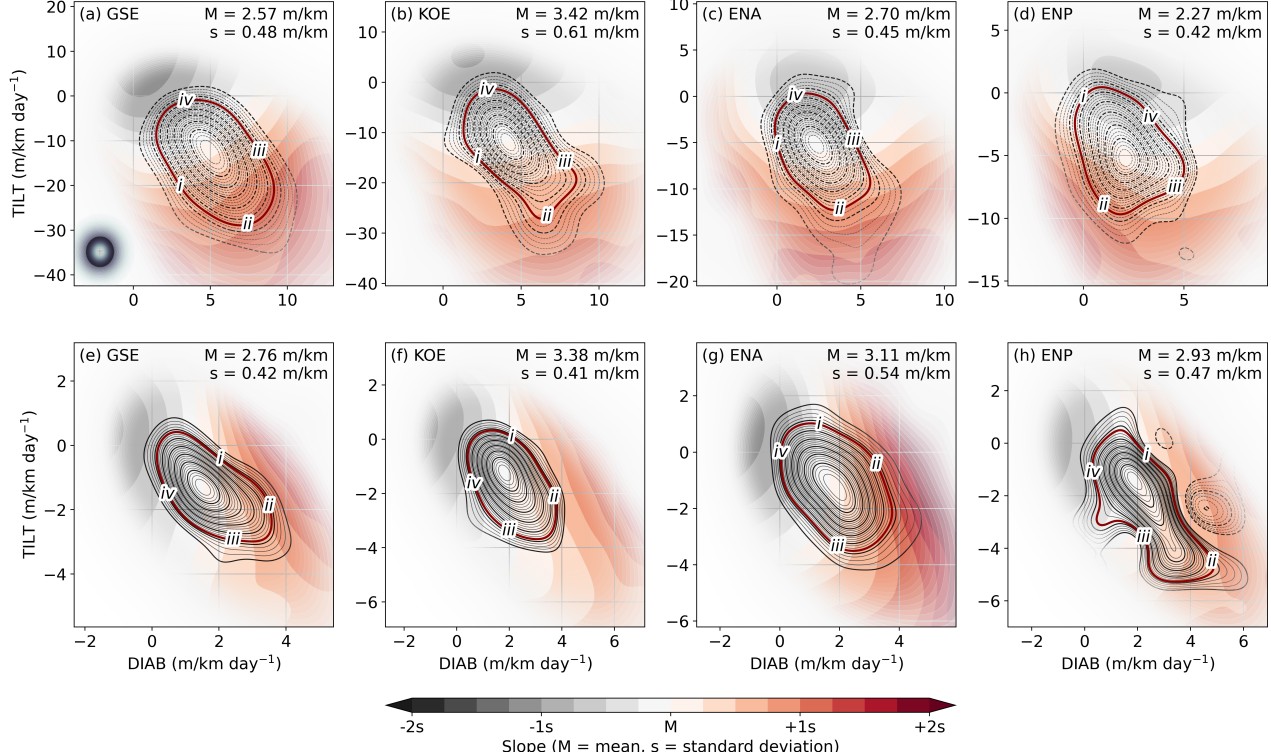

**Figure 3.** Phase portraits of spatially averaged DIAB (x-coordinate) and TILT (y-coordinate). Shading represents the kernel-averaged mean-slope, offset and scaled according to the mean and standard deviations of the slope time series, respectively, which are annotated in the upper right corner of each panel. Contours represent the stream functions associated with the kernel-averaged phase space circulation, with positive (solid) and negative (dashed) values indicating a clockwise and anticlockwise direction, respectively. Grid points are faded according to data density, such that kernel averages for points where data is scarce are blanked out. The upper panels show phase portraits for the near-surface GSE (a), KOE (b), ENA (c), and ENP (d) regions, while corresponding panels below show phase portraits for the free troposphere. The size of the Gaussian filter used to construct the phase portraits is indicated by the black-shaded dot in the lower-left corner of panel a, while the white dot within it represents that for the computation of kernel composites in Figs.5–10. Roman numbers (i, ii, iii, iv) indicate the four different points in the phase space where the kernel composites are evaluated (see Figs. 5–10).

170    double the range of variability downstream (Fig. 3c,d). In the free troposphere, however, there is no substantial difference in the magnitude of the average tendencies between upstream and downstream regions (Fig. 3e–h).

    The mean isentropic slope in the near-surface troposphere increases both with DIAB and TILT, reaching maximum values around one standard deviation above its time-mean in the lower-right quadrant of the phase space in the upstream sectors (Fig. 3a,b), while it increases primarily with TILT and peaks in the lower-left quadrant in the downstream sectors, especially

175    for KOE and ENP (Fig. 3c,d).





Given that the net effect of TILT on the slope is negative, it appears counter-intuitive that the near-surface slope increases with TILT. It is useful to reframe the causal link between slope and its tendencies, as in the near-surface it is more likely the case that the slope drives changes in TILT and DIAB, which in turn intensify primarily as the slope becomes steeper. That does not imply that slope is the only independent dynamical variable because TILT and DIAB are not solely dependent on slope.

In the free troposphere, on the other hand, DIAB leads on TILT and the area-mean slope increases primarily with DIAB in all of the domains considered (Fig. 3e–h). The interpretation is thus more straightforward, as the increase of slope with DIAB underlines the role of DIAB driving storm development, which in turn manifests itself in TILT.

## 5 Phase space composites

We can gather further insight into the mechanisms driving the phase space circulation by evaluating composites of the atmospheric flow at various locations in the phase space. Specifically, we identify four stages corresponding to (i) increasing slope, (ii) steepest slope, (iii) decreasing slope, and (iv) weakest slope. The specific location of each stage within the phase space is shown in Fig. 3 and the time intervals are indicated in Table 2. The exact location of each stage was determined so that it eases the comparison between the different regions. Consequently, time intervals between different stages are somewhat different, ranging between ≈0.7–2d. It should be noted that the time reference is used mostly for convenience in following the evolution along a trajectory and does not necessarily represent the typical duration of a stage.

We construct kernel composites of geopotential height at 1000hPa (Z1000) and 500hPa (Z500), as the spatial shift between these two levels is informative on the baroclinic structure of the atmospheric flow, where we removed the winter climatology (Fig. 4) to highlight transient variability. Even in the climatology, the midlatitude circulation features baroclinic growth with a westward tilt with height, as minima in Z1000 are located east of Z500 minima (Fig. 4).

We calculate composites of DIAB and TILT to visualise their spatial structure along a lifecycle, where we inspect full fields as opposed to anomalous fields, as the interpretation of anomalies for accumulated fields can be misleading. For the free troposphere, we also consider the full composites of total column water vapour (TCWV) and total precipitation (sum of large-scale and convective) to highlight the connection between high levels of moisture availability and the bulk of diabatic processes associated with latent heating.

### 5.1 Near-surface composites

The first stage is dominated by TILT, where all of the four domains show negative anomalies in Z500 and Z1000 (Fig. 6a,b and Fig. 5a,b), indicating advection of cold air from continents for KOE and GSE and from polar oceans over warmer ocean surfaces for ENP and ENA. The structure of the flow is consistent with the onset of CAOs, which are most frequent in these regions in winter (Grønås and Kvamstø, 1995; Dorman et al., 2004; Kolstad et al., 2009). TILT follows the advancing cold air front, while DIAB intensifies further upstream.

The second stage is characterised by the steepest slope and coincides with maximum TILT while DIAB is still increasing. Composites for this stage (Fig. 6c,d and Fig. 5c,d) show a strengthening and, especially in the ENA region (Fig. 6c), a down-



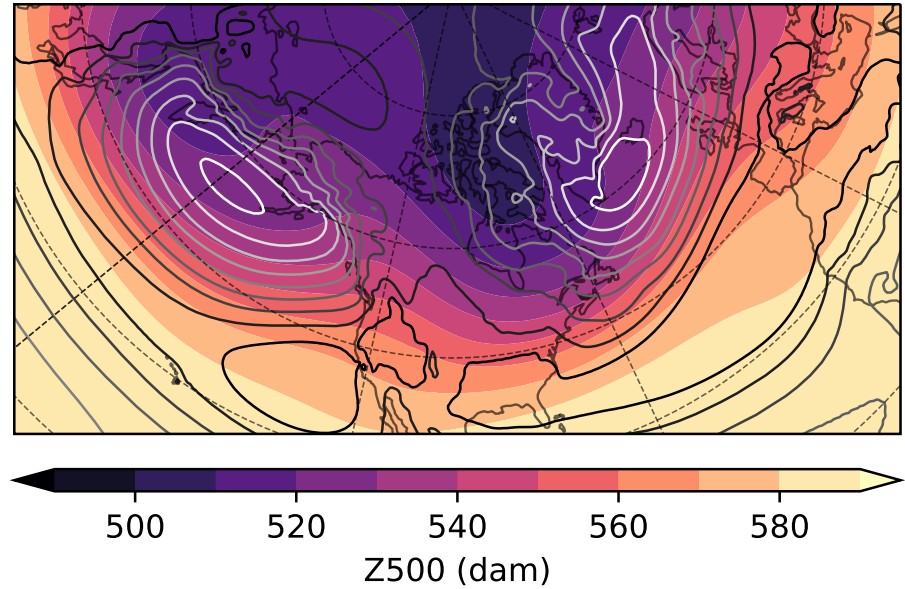

**Figure 4.** Winter climatology (NDJF, 1979–2017) of geopotential height at 500hPa (Z500, shading) and 1000hPa (Z1000, contours, every 2dam between 0–15dam, whiter contours for lower geopotential).

stream progression of the cyclonic circulation that emerged in the previous stage. TILT again spatially precedes DIAB, which is now dominant and gradually spreads downstream. For ENA and ENP (Fig. 6c,d), the westward tilt in geopotential becomes

weaker compared to GSE and KOE (Fig. 5c,d), suggesting that the cyclonic anomalies downstream are in a more mature, barotropic stage. The associated atmospheric flow is, however, still favourable for CAOs, with northwesterly flow from the continents and northerly flow from the polar oceans.

In the third stage (Fig. 6e,f and Fig. 5e,f), TILT is steadily subsiding while, especially along the western boundary currents, DIAB prevails (Fig. 5e,f). CAOs are no longer active within the domains, as the cyclonic circulation gradually weakens and

positive, anticyclonic geopotential anomalies start building up in the western part of the domains.

The anticyclonic circulation emerging in the third stage becomes a dominant feature of the fourth and final stage, where TILT and DIAB as well as the mean slope are weakest (Fig. 6g,h and Fig. 5g,h). The anticyclonic anomalies mainly represent the suppression of cyclonic activity, consistent with a weaker slope. The vertical structure is still baroclinic in the upstream regions (Fig. 5g,h) while it is more barotropic downstream (Fig. 6g,h), reflecting the climatological structure of the atmospheric

flow in these regions.

## 5.2   Free-troposphere composites

In the free-troposphere, slope across the phase space concurs to a greater degree with changes in DIAB than TILT. In the first stage, slope is slowly building up as cyclonic geopotential height anomalies are forming either within (GSE and KOE,



Fig. 7a,b respectively) or slightly to the west of the spatial domain (ENA and ENP, Fig. 8a,b respectively). DIAB is most active
in close proximity to these cyclonic anomalies, particularly in correspondence with the upper-level anomalies (Z500), while
TILT remains almost negligible.

The cyclonic anomalies deepen further in the second stage, especially in the GSE (Fig. 7c), ENA (Fig. 8c), and ENP (Fig. 8d)
regions, while a similar but weaker development is evident in KOE (Fig. 7d). Positive, anticyclonic anomalies are also forming
to the east of the cyclonic anomalies, while a second negative anomaly develops further eastward in the ENP composite
(Fig. 8d). The area of most intense DIAB broadens substantially, almost spanning the entire domains in KOE and ENP (Fig. 7d
and Fig. 8d) while TILT also increases in correspondence to the cyclonic anomalies and within areas of intense DIAB.

The third stage is characterised by a breakdown of the cyclone-anticyclone system, with diminishing amplitudes of the
geopotential height anomalies in the composites for GSE, ENA, and ENP (Fig. 7e, Fig. 8e,f) also acquiring a more barotropic
structure, especially in ENA and ENP. The evolution in KOE (Fig. 7f) is different from the other regions, as baroclinic,
anticyclonic anomalies become dominant across the domain, with traces of the cyclonic anomalies waning to the west. In
all four domains, DIAB subsides while TILT is still relatively strong. Mean slope is also decreasing, which, coupled with
stronger TILT, hinders further synoptic development.

In the fourth stage, slope has reached its minimum value and positive geopotential anomalies dominate in all domains.
Both DIAB and TILT are weak, with the only visible traces outside of the domains and mostly linked to their climatological
distribution (Fig. 1d,f).

Both the steepest slope and highest DIAB occur in the second stage when the cyclonic activity that developed in the first
stage reaches its maximum intensity. The pattern resembles a propagating Rossby wave stretching across the ocean basins,
which is most evident for GSE (Fig. 7c) and ENP (Fig. 8d) and to a lesser extent for KOE (Fig. 7d) and ENA (Fig. 8c). In all
composites, Z1000 and Z500 anomalies at stages (ii) and (iv) share a similar spatial pattern but with opposite sign, while stages
(i) and (iii) reflect the transition between these opposing 'phases' of a Rossby wave. Our results thus indicate that diabatic
processes constitute a defining feature of the evolution of wave activity across the midlatitudes.

Recent studies have shown that precipitation in climate models can have a significant impact on the representation of storm
tracks and jet variability (Schemm, 2023; Fuchs et al., 2023). However, the mechanisms that explain these sensitivities in
modelled storm track variability remain unclear. As latent heating associated with precipitation constitutes the bulk of diabatic
processes in the free troposphere (Papritz and Spengler, 2015), we can gather further insight into the role of moisture in storm
tracks dynamics by looking into the evolution of TCWV and total precipitation in the TILT-DIAB phase space.

Across all four regions, the most intense precipitation occurs during the second stage (panels c,d in Figs. 9–10) and, to a
lesser extent, during the third stage (panels e,f in the same figures). Although differences in the distribution of TCWV from
one stage to the other can be subtle, they are consistent with composites of precipitation and DIAB. In particular, we find
spatial correspondence between maxima in precipitation and strongest DIAB (darker shading in Figs. 9–10), which highlight
the primary importance of latent heat release in the diabatic restoration of slope.

Most of the precipitation appears to be located along trailing cold fronts associated with the strongest cyclonic anomalies
(panels c–f in Figs. 9–8), while much weaker signals emerge during stages dominated by anticyclonic anomalies (panels a,b





and g,h in Figs. 9–8). There is a reduction in TCWV behind the cold front, which is linked to the relatively dry air mass

associated with the cold sector. Some of the precipitation is also found to the east of the cyclonic systems (most clearly for GSE, KOE, and ENP), in correspondence with their warm sectors where TCWV is larger.

   The close relationship between DIAB, precipitation, and TCWV pinpoints the crucial role of moisture availability and moist diabatic processes in the compound evolution of cyclones, as previously suggested by Weijenborg and Spengler (2020), which is most likely linked to pulses in storm track activity.

**6   Conclusions**

We find that TILT and DIAB are strongly tied to each other throughout the troposphere, both in space (Fig. 1) and time (Fig. 2). Conducting a phase space analysis, we reveal a phasing between DIAB and TILT, which is of opposite sign in the near-surface (900-800hPa) and free (750–350 hPa) troposphere (Fig. 3), where TILT and DIAB lead, respectively.

   For the near-surface troposphere, TILT leads DIAB (Fig. 3a-d), associated with the advection of cold air masses during

CAOs over the warmer ocean, which initially yields a flattening of isentropic surfaces followed by a response in surface heat fluxes upstream that restores the slope (Figs. 5,6). This evolution goes hand in hand with both DIAB and TILT, but does not appear to be driven by neither DIAB nor TILT alone. Instead, the steepest slope is found when the magnitude of both TILT and DIAB are largest, which supports the idea that changes in slope actually condition DIAB and TILT, thus driving the circulation in the phase space.

Phase portraits for the free troposphere, on the other hand, point to the primary role of DIAB driving the phase space circulation, as DIAB leads in time on TILT, while their intensification goes hand in hand with steepening of the slope (Fig. 3e-h). Composites across the phase space confirm that DIAB takes place ahead of storm activity, both in time and space (Figs. 7,8). In particular, composite analysis reveals that the evolution across the phase space is part of a Rossby wave packet stretching over the oceanic basins, suggesting that DIAB in the free troposphere is a distinctive aspect of the evolution of storm tracks,

while the development and progression of anomalies in the composites for the near-surface are more contingent to the specific spatial domain considered.

   Furthermore, we find that most of DIAB coincides with peaks in precipitation and overall moisture availability (Figs. 9,10), further underlining the differences that sensible and latent heating play in the lower and upper troposphere, respectively, as pointed out in previous studies (e.g., Hoskins and Valdes, 1990; Hotta and Nakamura, 2011; Papritz and Spengler, 2015). The

close link between free-tropospheric DIAB and moisture availability adds to a growing body of literature highlighting the importance of moist-diabatic processes and their correct representation in numerical models to reduce model biases in storm track location and intensity (Willison et al., 2013; Schemm, 2023; Fuchs et al., 2023) and pinpoint the need to develop a more comprehensive moist storm track model.



*Data availability.* Data from ECMWF ERA-Interim (Dee et al., 2011) are freely available from the ECMWF at https://www.ecmwf.int/en/
forecasts/dataset/ecmwf-reanalysis-interim (last access: June 2023)

*Author contributions.* AM performed data analyses and prepared the paper. TS contributed to the interpretation of the results and to the writing of the paper.

*Competing interests.* The authors declare that they have no conflict of interest.

*Acknowledgements.* We would like to thank the ECMWF for freely providing Reanalysis data, and Clio Michel for making available cyclone
tracks data. All slope diagnostics were calculated using tools from the Python library *Dynlib* (Spensberger, 2021). This study was supported by the Research Council of Norway (Norges Forskningsråd, NFR) through the BALMCAST project (NFR grant number 324081).



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

375

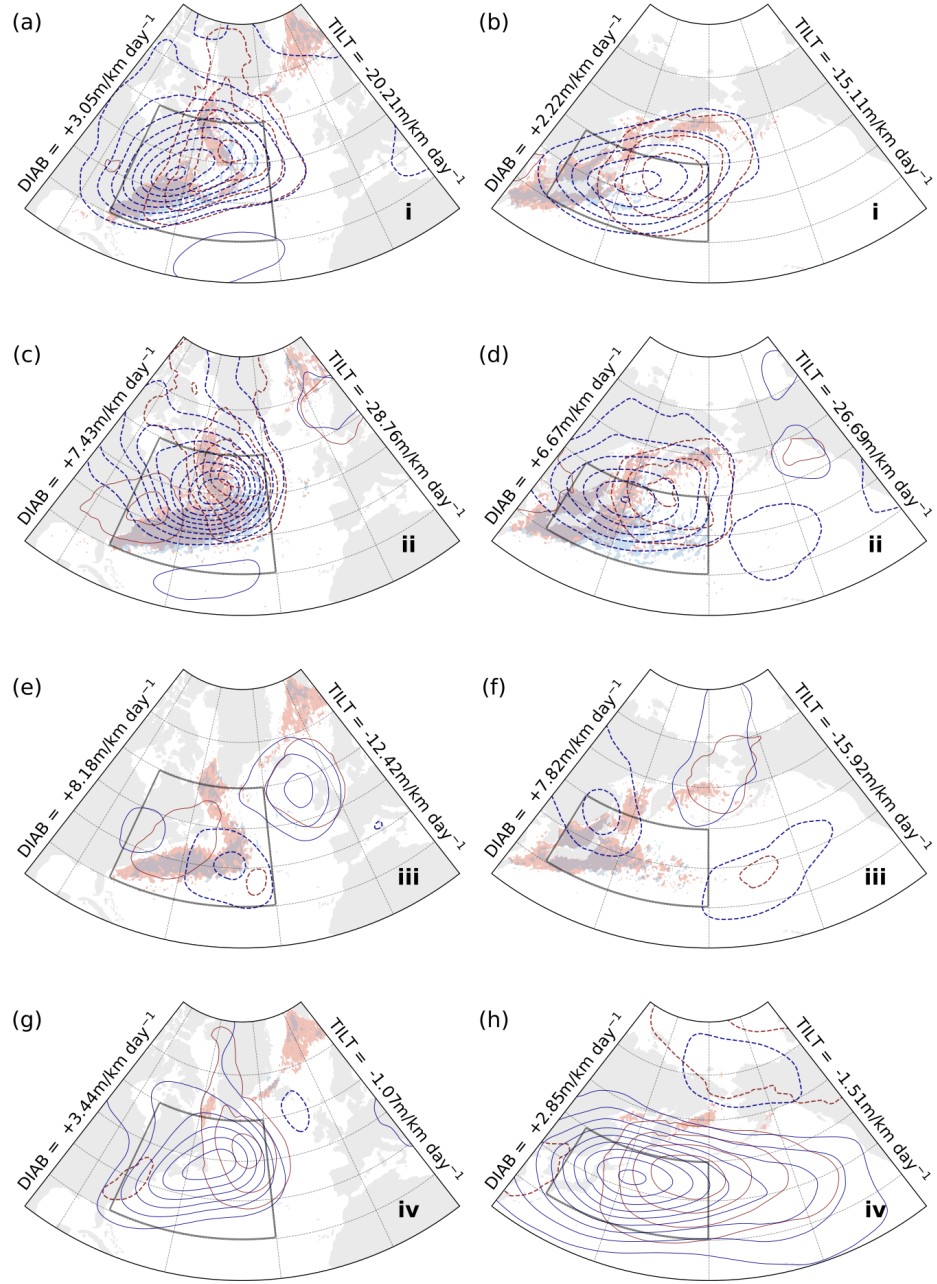

**Figure 5.** Kernel-averaged composites of Z1000, Z500, DIAB, and TILT for (a,b) phases i, (c,d) ii, (e,f) iii, and (g,h) iv in the phase portraits for the near-surface troposphere upstream regions GSE (left, Fig. 3a) and KOE (right, Fig. 3b). Contours of Z1000 and Z500 (red and blue, respectively) represent the anomaly field relative to climatology (NDJF, 1979–2017) and are plotted every 2 dam (0 dam contours omitted, negative contours dashed). Shading for DIAB (red) and TILT (blue) represents full composite (not anomalies) and indicates values above 15 m/km day$^{-1}$.



**Figure 6.** Same as in Fig. 5, but for the near-surface troposphere downstream regions ENA (left, Fig. 3c) and ENP (right, Fig. 3d). Shading for DIAB (red) and TILT (blue) indicates values above 6 m/km day$^{-1}$.





**Figure 7.** Same as in Fig. 5, but for the free-troposphere upstream regions GSE (left, Fig. 3e) and KOE (right, Fig. 3f). Shading for DIAB (red) and TILT (blue) indicates values above 3 m/km day$^{-1}$ in magnitude.



**Figure 8.** Same as in Fig. 5, but for the free-troposphere downstream regions ENA (left, Fig. 3g) and ENP (right, Fig. 3h). Shading for DIAB (red) and TILT (blue) indicates values above 3 m/km day$^{-1}$ in magnitude.

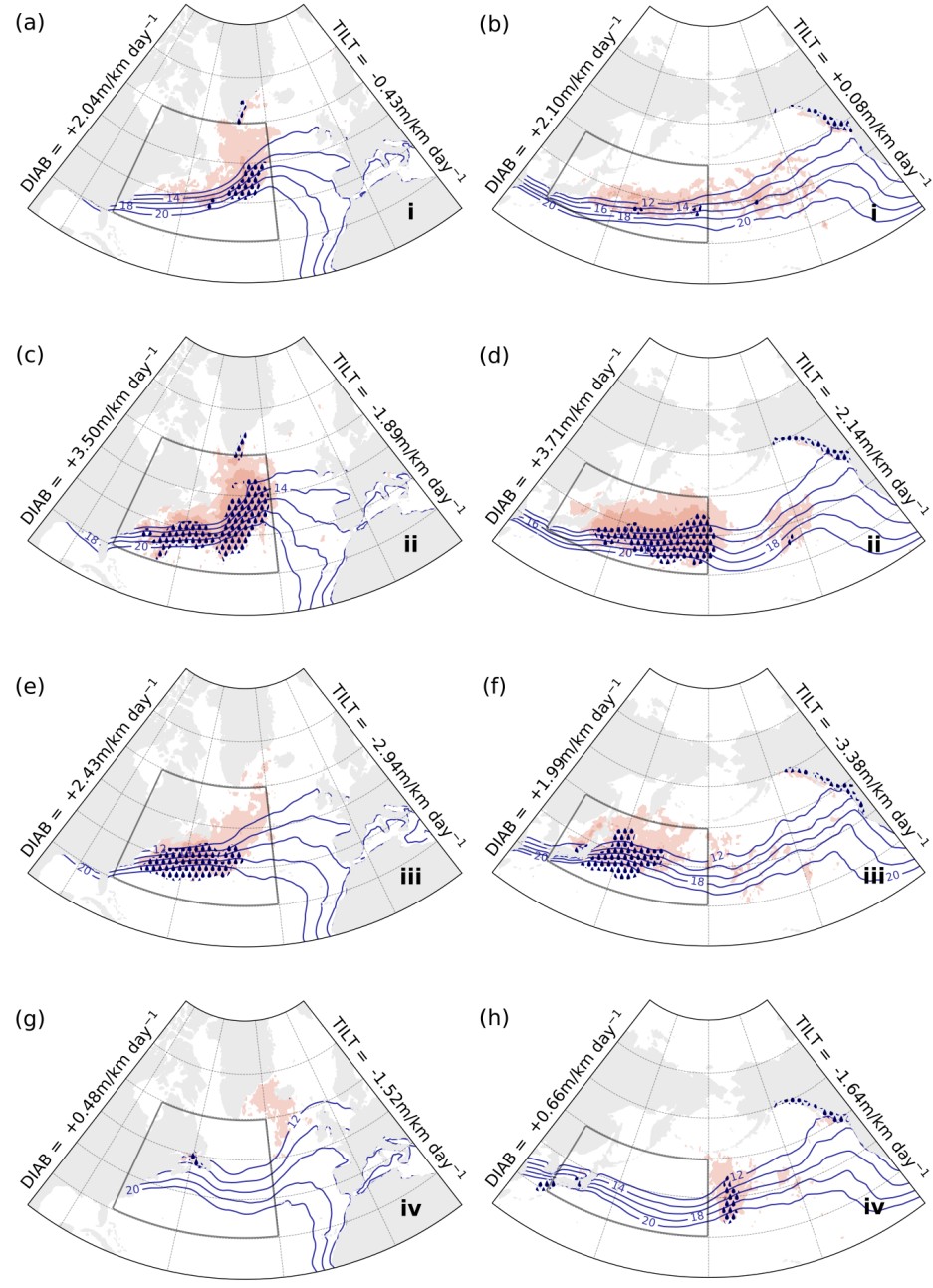

**Figure 9.** Kernel-averaged composites of DIAB (red shading, lighter above 3 m/km day$^{-1}$, darker above 5 m/km day$^{-1}$), TCWV (blue contours, every 2 kg m$^{-2}$ between 12–20 kg m$^{-2}$) and total precipitation (hatching, above 8 mm day$^{-1}$) for phases i (a,b), ii (c,d), iii (e,f), and iv (g,h) in the phase portraits for the free-troposphere upstream regions GSE (left, Fig. 3e) and KOE (right, Fig. 3f).



**Figure 10.** As in Fig. 9, here for the free-troposphere downstream regions ENA (left panels, Fig. 3g) and ENP (right panels, Fig. 3h).