# Peer review of "Diabatic effects on the evolution of storm tracks"

_EGUsphere, 2023_

## Author Comment (AC1)

**Response to Maarten Ambaum**

This work addresses an important problem in atmospheric dynamics, namely the dynamical processes underlying the formation of the baroclinic zone, as diagnosed by the slope of isentropic surfaces. A new perspective is introduced by looking at the slope evolution in phase space. It is found that the upper and the lower troposphere exhibit very distinct (in some sense opposite) dynamics for the slope evolution.

My own work also looks at phase space dynamics of the storm track and it will be no surprise that I find this approach of great interest. However, the manuscript left me wondering in the end, what the real conclusion was regarding the role of diabatic effects in slope dynamics. The writing is somewhat evasive and vague at many points so it is hard to interpret what the authors are actually saying.

I would imagine that a more clearer layout of the arguments at those points should clarify sufficiently what was specifically meant. Below I comment on the manuscript in order of line number, not in order of importance.

We thank Maarten for such a rapid and thorough review of our work. We hope to have addressed all the concerns raised, which helped us improve the readability of the manuscript. Here we provide a line-by-line response.

**Specific comments**

1) l.12: "... rendering the phasing ..."; did you mean something like "... suggesting that this phasing is ..."; I also assume that "phasing" actually means the order in which processes occur; perhaps that would be a clearer and simpler way of putting it?

   Our use of the verb render is perhaps not very common. To improve readability, we rephrased as follows: "The same phasing between diabatic and tilting tendencies of the slope is observed both in upstream and downstream sectors of the North Atlantic and North Pacific storm tracks. This suggests that the opposite behaviour between near-surface and free troposphere is a general feature of midlatitude storm tracks."

2) **l.88 Presumably you meant** $d\theta/dz < 10^{-4}$... **not** $> 10^{-4}$

Yes, thanks for spotting the typo. We have fixed it.

3) **l.92:"... does not affect our analysis as we exclude land grid points." Given that the cold air for the all-important cold air outbreaks is sourced on the land, it seems to me that it is strange to exclude land points: as a local tendency it is perhaps not important, but as a source of required air masses it is. How is that reflected in your equations or your analysis? It seems to me that the anomalously high diabatic cooling over land is absolutely crucial in providing a source of the gradient in the baroclinic zone.**

The isentropic slope and its tendencies are computed everywhere and we then exclude land grid points when computing spatial averages. So, the contribution of diabatic processes over land enters the domain of interest over the ocean through the slope entering the domain. We exclude land grid points to avoid the signature of orographic effects, which can dominate the response in TILT and thereby confuse the analysis. We have edited the manuscript to make this more explicit. "Over land, the largest amount of masking occurs in correspondence with high orography, though this does not affect our analysis as we exclude land grid points when calculating spatial averages."

4) **Fig 1: Perhaps label the figures which are free troposphere and which are lower troposphere? Also: you use the word "hatching" in this figure and figure 9, but you actually show a stippling. (I thought I'd check the dictionary on my computer, just to be sure. This is the entry: "hatching | hatʃɪŋ | noun [mass noun] (in fine art and technical drawing) shading with closely drawn parallel lines: the miniaturist's use of hatching and stippling.")**

We have labelled the figures accordingly, explicitly mentioning the pressure levels between which the vertical average is taken (we realised we actually meant 900-825hPa for the near-surface, so we have changed that in the text as well). Thanks for pointing out the difference between stippling and hatching,

one never stops learning! In our defence, the relevant command in Python uses 'hatches' for both, still two wrongs don't make a right, so we have corrected this in the revised manuscript.

5) **l.132: "... mainly due to orographic effects advected ..." I do not know what the process is that you are describing. Please be specific what you actually mean. How does orography swap the sign of these effects, and how does this advection work?**

By orographic effects, we refer to gravity waves and other mesoscale features excited by mountain ranges that can result in significant TILT. At instances when TILT is generally weak over most of the spatial domain, these orographic effects can dominate the domain-averaged value for the tilting term. To reduce the impact of these undesired orographic effects, we adjusted the domains accordingly. We have rephrased the manuscript as follows to clarify this point:

"There are instances where they change sign, most often for TILT rather than DIAB. This is mainly due to orographic effects (e.g., gravity waves and other mesoscale features excited by mountain ranges) advected over the ocean that can result in significant positive TILT and thus dominate its domain-averaged value, especially when TILT is generally weak. To reduce the impact of these undesired orographic effects, we adjusted the domains accordingly."

6) **Fig.2: Looking at the lower panel, it would be of interest to add a third line indicating "DIAB + TILT"; it is after all the lack of compensation between those two terms that provides the tendency for the slope. In that respect, could you please comment, perhaps somewhere around l. 135, whether the IADV term is smaller than the DIAB + TILT combination? If DIAB and TILT are largely compensating each other then their individual magnitudes do not matter that much. It seems to me that the mismatch between DIAB and TILT is the crucial variable. (Of course, the phase space analysis is diagnosing that mismatch in detail.)**

We have looked in more detail into the discrepancy between DIAB and TILT.

Figure AR1 is an extended version of the manuscript's Fig. 2, which also includes time series for DIAB+TILT (Fig. AR1c) and for the advection term, ADV (Fig. AR1d). We show ADV instead of IADV as we have reformulated the slope tendency equation (Equation 2) in its Eulerian form rather than Lagrangian, in response to the other Reviewer. In the Eulerian form, the advection term also includes local slope tendency due to the adiabatic advection of slope by the flow.

We notice that the sum of DIAB and TILT tends to be on the positive side, although it does oscillate considerably. In particular, positive (negative) values appear to correspond to a higher (lower) slope (Fig. AR1a), though it remains somewhat hard to link visually. The magnitude of the advection term is substantially smaller than DIAB and TILT, while it is more comparable to that of DIAB+TILT but still visibly weaker. As we deem it an interesting result in itself, we have decided to comment upon it in the revised manuscript, including the following lines on line 135 of the original manuscript:

"We also notice that the sum of DIAB and TILT tends to be on the positive side (not shown), with positive (negative) values appearing to correspond to a higher (lower) slope. While the advection term is substantially weaker than DIAB and TILT (not shown), its magnitude is comparable to that of DIAB+TILT, which suggests that part of the imbalance between DIAB and TILT is compensated by advection."

However, we have not included DIAB+TILT in the revised manuscript as it does not really add much to Figure 2 and, while it is true the mismatch is expected to be more relevant to changes in slope, here we want to focus on the phasing between DIAB and TILT and the physical mechanisms behind it.

**7) Fig.3: This is a crucial figure, but it is also very dense and hard to read. Some separate comments:**

**It feels more natural to me to put the free troposphere images at the top. I do not see any problem with describing the bottom plots first, Sn. 5.1, then the top plots, Sn 5.2.**

**Could you add a couple of arrows, perhaps to Figs 3a and 3e, to indicate the direction of flow? Could you add a 1-1 line (or rather a line with slope -1) to indicate the line in phase space where TILT and DIAB exactly compensate and will not contribute to slope**

[Figure]

Figure AR1: Time series of free-tropospheric (a) isentropic slope, (b) DIAB and TILT, (c) DIAB+TILT, and (d) ADV, all spatially averaged over the GSE region. Solid lines represent a sample season (NDJF 2013-14) while light (dark) shading represents the interdecile (interquartile) ranges over the climatological period (1979–2017).

**tendency. In this sense, phases (i) and (iii) are defined by the maximum distance from this line, and perhaps phases (ii) and (iv) need to be defined as lying exactly on this line.**

We have inverted panels so that the free troposphere and near-surface phase

portraits are shown in the upper and lower panels, respectively. We have also added arrows to the leftmost panels to illustrate the direction of the phase space circulation more clearly.

A line indicating where TILT and DIAB compensate is not particularly useful, especially in the near-surface where TILT is considerably stronger than DIAB. At an earlier point, we envisaged using a more objective method to determine the four different stages. In particular, we opted to define them by looking at the value of the kernel-averaged slope: increasing, maximum, decreasing, and minimum. However, the transition from one stage to the next was somewhat harder to follow and consequently required a larger number of panels to be able to link them. That by itself is perhaps indicative that such 'analytical' and elegant definitions do not always correspond to the most relevant points in the phase space.

To summarise our results efficiently and concisely, we focused on the existing sets of four points in the phase space, whose selection aims to enhance the visualisation of the transitions between distinct stages.

**8) l.176-179: I found this a very confusing bit. Are you describing your phase space portraits or are you actually speculating on mechanisms? What do you mean by "driving": a description of what happens in the figures, or some causal mechanism? If a physical mechanism, you would need to be more specific: what processes are actually happening? If describing the figure, perhaps use a different word to "driving". Perhaps "leading"?**

Here we describe the phase portraits and highlight one aspect that appears counter-intuitive at first, as we observe mean slope to increase while TILT is stronger than DIAB, which would be expected to coincide with a decreasing slope. One way to resolve this apparent contradiction is to re-frame the causal link between slope and TILT, namely that a steeper slope could enhance TILT and DIAB. However, at this stage, this is simply a speculation, as we cannot say much more by looking at the phase portraits alone. Composite analysis indicates that the advection of cold air over the ocean corresponds with a surge in TILT and DIAB, so the advection likely plays a more relevant role in the near-surface. We have expanded the discussion on lines 176-179 of the original manuscript to make this point clearer.

"Given that the net effect of TILT on the slope is negative, it might at first appear counter-intuitive that the near-surface slope increases with TILT. One

possible explanation for this apparent contradiction is that the steepening of mean slope actively contributes to the strengthening of both TILT and DIAB. Using phase space composites, we shed light onto the physical mechanisms behind the phase space circulation (see Section 5)."

9) **l.182: "...driving storm development": DIAB is a tendency for isentropic slopes, not for storm development.**

We wanted to stress the leading role that DIAB plays in the free troposphere, leading in time both on slope and TILT. The steeper the slope, the more favourable the conditions for storm development, which is captured by TILT. We have rephrased the text as follows: "The interpretation is thus more straightforward, as the increase of slope with DIAB underlines the primary role of DIAB in generating enough slope for the development of baroclinic instabilities, which in turn is associated with an increase in TILT."

10) **l.200: Just a personal comment regarding style which, obviously, you may completely ignore. This paper is not easy to follow because of all the shorthand notation: TILT DIAB IADV KOE GSE ENP ENA TCWV CAO, . . . Please consider writing this stuff out in the text: there is no gain in shortening it, and it is a real pain to follow the text when using all this shorthand. I know it is a device used by many authors, but I believe that a paper should be as easy as possible to follow; Mike Mcintyre would agree with me on that. Do you really want readers to be trying to decipher what a CAO over the KOE is?**

While we agree with the Reviewer's sentiment on this point, we argue that there is a gain in introducing the shorthand notations used in this paper and their removal might make some paragraphs unnecessarily wordy or unclear.

For instance, the use of geographical tags helps us distinguish between the different domains, especially when describing composites: we might be referring to the Eastern North Pacific both in KOE and ENP composites. Hence we decided to retain GSE, KOE, ENA and ENP, which are presented in Table 1 so that it should be easier to find their definition.

As for CAO and TCWV, they are quite common abbreviations. In particular, TCWV is the shorthand notation used in ECMWF documentation, so it is obvious we are using this specific abbreviation.

Finally, DIAB and TILT are easier to refer to compared to diabatic and tilting tendencies, especially when we mention increases/decreases of such tendencies.

11) **l.204: "TILT follows the advancing cold air front, while DIAB intensifies further upstream.": It really would help if you explained these things a bit more specifically: why/how does TILT "follow" the advancing cold air? It is often not clear whether you are trying to describe the plots or to explain the plots. Could a schematic help?**

Here we were describing the spatial distribution of TILT and DIAB in the composites and, specifically, referring to the fact that at this stage tilting tendencies (TILT) strengthen in correspondence with the advancement of cold air masses over the ocean, which is consistent with the circulation associated with geopotential anomalies. To clarify this point we have rephrased the text as follows: "The spatial distribution of strong TILT follows the advancing cold air front, while DIAB intensifies upstream to the west of peaks in TILT."

12) **And I remain somewhat puzzled: if "TILT follows the advancing cold air front" would this then not be manifested in the IADV term? I probably misinterpret the physical processes underlying these two terms, but it surely is an indication that it is not obvious what you are actually saying here.**

The isentropic advection term (IADV) represents along-flow changes in the slope of an air parcel. In the absence of processes that actively modify the slope (either diabatically or adiabatically), the slope of the parcel would change as it moves into an environment with a different slope. The advection of a cold air mass concurs with physical mechanisms that deform isentropic surfaces, thus TILT, or actually its spatial distribution, can 'follow' the cold front and not be captured by the advection term. In an Eulerian perspective,

the local slope tendency is also affected by the advection of slope (Papritz and Spengler,2015). However, by definition, TILT would still exclude advection and only measure the tilting contribution to the local tendency. To avoid confusion, we have rephrased the text to avoid using the term 'follows': "The spatial distribution of strong TILT trails behind the advancing cold air front, while DIAB intensifies upstream of peaks in TILT (i.e., to their west)."

13) **l.218: "... the suppression of cyclonic activity, consistent with a weaker slope.": I am not sure what you are saying here: a weaker slope corresponds to reduced baroclinic growth. This is not obviously the same as saying that a suppressed cyclonic activity is consistent with a weaker slope. What do you mean with "suppression" anyway? Perhaps you simply meant "represent the anomalously low cyclonic activity. . . "?**

The use of the term 'suppression' was perhaps a bit misleading so we have rephrased as suggested.

14) **l.245: What is the purpose of the hyphens around "phases" Are they the opposing phases of the Rossby wave or not? (I think they are.)**

We have removed the hyphens.

15) **l.246: "... constitute a defining feature ..." : I do not know what this means. Can you be more specific? In what sense is it a "defining" feature? With this phrase it certainly doesn't sound like you managed to add physical understanding to the role of diabatic effects in the evolution of the storm track, which is probably underselling the results presented.**

As we are able to capture the evolution of a Rossby wave packet across the North Atlantic and Pacific oceans through composites based exclusively on changes in DIAB and TILT, our results suggest that the phasing between

DIAB and TILT, with DIAB leading TILT, characterises the propagation of such Rossby wave packets. Perhaps our use of 'defining' here is not generally intelligible, so we rephrased it as follows to make it more explicit. "These composites are based exclusively on the mean value of DIAB and TILT, which implies that their evolution is inherently linked to how DIAB and TILT co-evolve. Through composites at different stages, we are able to reconstruct the propagation of a Rossby wave across the North Atlantic and Pacific oceans, and the specific phasing between DIAB and TILT, with DIAB leading TILT, appears essential in its propagation."

**16) l.262: "... of moisture availability": Moisture availability seems to imply that DIAB is a moisture limited process. Do you have evidence for that, or do you mean something else?**

In the same section, we show how peaks in DIAB correspond to an increase in total column water vapour, which suggests that moist diabatic processes underpin a substantial fraction of the diabatic tendency. However, from re-analysis alone, we cannot determine whether moisture is actually a limiting factor, which is a hypothesis that would require sensitivity experiments to validate. We have made it clearer in the revised manuscript that our results are consistent with such a hypothesis, though further work would be needed. Specifically, we rephrased lines 262–264 as follows: "The close relationship that we found between DIAB, precipitation, and TCWV is consistent with the hypothesis that moisture availability plays a crucial role in the evolution of cyclones and is most likely linked to pulses in storm track activity, as suggested by Weijenborg and Spengler (2020). However, sensitivity experiments are needed to validate this hypothesis."

We also added a line at the end of the conclusions, which points more explicitly to possible future work.

**17) l.272: "neither": Double negative; it needs to be "either".**

We have corrected this in the revised manuscript.

**18) l.273: "... actually condition DIAB and TILT": what do you mean with the verb "condition"?**

We refer back to the dictionary entry for condition (verb): *have a significant influence on or determine (the manner or outcome of something)*, so here we mean that changes in the near-surface slope lead to changes in DIAB and TILT rather than the other way round.

**19) l.275: " driving": "Driving" or "leading" (as in "leading in time")? Driving seems to imply a causal effect; if so, you need to explain how this causality works. The rest of the sentence seems to just describe the figure, but I think you developed a much clearer physical picture of why DIAB leads TILT in this case.**

Here we meant to briefly summarise the main results for the free troposphere, namely that phase portraits indicate that the phasing between DIAB and TILT suggests the former leads on the latter. By itself, this result would not help infer any causality, which arguably would make the use of the term 'driving' somewhat speculative at this stage. However, when we inspect composites in the phase space, we are able to visualise the typical structure of the atmospheric flow at different points in the phase space. These composites suggest that the initial increase in DIAB occurs primarily to the south/southwest of the cyclonic anomalies that will then deepen and become part of a wave packet stretching along the oceanic basins. The diabatic generation of slope on the downstream side of these cyclonic anomalies contributes to the increase in mean slope which then fuels their further development and triggers TILT as these cyclones consume more and more of the excess baroclinicity/slope.

We agree the original wording was not as clear as it could have been and we have rephrased the entire paragraph in light of this and further comments below, also by the second Reviewer.

**20) l.279-281 "... suggesting that DIAB in the free troposphere is a distinctive aspect of the evolution of storm tracks, while the development and progression of anomalies in the composites for the near-surface are more contingent to the specific spatial domain**

considered.": There are many words here, but it is not obvious what is actually said. Most people would agree, without any hesitation, that diabatic effects play an important part in the evolution of storm tracks (I am not sure what a "distinct aspect" is –distinct from what?). I also do not understand the final part of this very long sentence. What are you trying to say?

We acknowledge this sentence was poorly phrased. We have rephrased it to link it back to the discussion already presented in Section 5.2.

"In particular, the evolution of the anomalous circulation (Z1000 and Z500) across the different stages identified in the phase portraits is reminiscent of a Rossby wave packet propagating over the North Atlantic and Pacific Ocean basins. As composites are based exclusively on changes in DIAB and TILT, the specific phasing between DIAB and TILT appears thus an essential feature of these Rossby wave packets."

21) To conclude, after reading this, it is not obvious what the main take-home message is of the paper. I think it is the distinct dynamics between the two altitudes. My take on it is that TILT and DIAB mostly compensate on average, but at lower levels TILT leads DIAB due to the low level cold air advection being followed by sensible heat fluxes, while at higher levels DIAB leads TILT because the forced latent heat release produces some counteracting secondary circulation. Perhaps I misread the gist of the paper, but I think that it should be more clearly laid out what this take-home message is.

We have partly rephrased and expanded the concluding section, hopefully laying out more clearly the main results of our study, which include the conclusions already mentioned by the Reviewer.

---

## Author Comment (AC2)

**Response to Reviewer N.2**

**This study combines a quantitative assessment of the main contributors in the tendency equation of isentropic slopes with a phase space analysis that is able to shed light on the joint temporal evolution of these factors. While both methods have been applied before, there combination yields novel insights into the temporal behavior of storm tracks, making this paper a valuable contribution to the literature that also fits well in the scope of Weather and Climate Dynamics. However, I think that there are still some weaknesses in the presentation of the results that I'd ask the authors to address before I can recommend the paper for publication. On the one hand, these are related to a somewhat superficial description of details in the figures, leading to a few unclear points listed below. On the other hand, at some places (in my view in particular in section 5), the general conclusions obtained from the analyses are not articulated clearly enough, as also noted by the other reviewer.**

We thank the Reviewer for their thorough review of our paper. We hope to have addressed the concerns raised, which also helped us improve the manuscript. Here we provide a line-by-line response.

**Specific comments**

1) **Line 58: Is there a reason why you still use the old ERA-Interim reanalysis instead of ERA5, which has been out for a few years now?**

    We used ERA-Interim, as these were readily available to us, in particular the diabatic tendency of the slope. To calculate the diabatic tendency we need diabatic tendencies of potential temperature on pressure levels, which the ECMWF does not provide directly but only on model levels. The process of requesting and converting the necessary data is quite slow (in the end it took 8 months), so we used ERA-Interim in the meantime. As synoptic-scale dynamics are unlikely to change significantly between the two datasets, we do not expect significant differences in our results. Future work will be based on ERA5 and preliminary results confirm the consistency with this study.

**2) Equation 2, line 85: This is more a conceptual (and also minor) point, but is it really the material derivative of S that you're aiming at, or rather the local tendency at a specific grid point (e.g., in your composite analysis)? Of course, this would not change your analysis at all, but it would rather mean that you neglect the ADV term in equation 16 of Papritz and Spengler (2015) than the IADV term in your equation 2, right?**

To compute the slope and its tendencies on pressure levels we made use of the Eulerian form of the tendency equation (namely, Equation 16 in Papritz and Spengler, 2015), where the advection term also includes advection of slope by the flow. As the Reviewer pointed out already, TILT and DIAB do not change between the Lagrangian and Eulerian formulations so our analysis is not affected. We do, however, agree with the Reviewer, that conceptually we are more interested in the local slope tendency and so it is more appropriate to show its Eulerian formulation. We have thus changed Equation 2 accordingly.

**3) L 90: "over the western boundary currents": This is true for the Pacific, but not for the Atlantic**

A previous version of Figure 1 showed stippling for masking occurring more that 15% of the time, which would make the Gulf Stream also pop up, hence our comment on Line 90. We have rephrased the text as follows to make the description more consistent; "The largest amount of masking over the oceans occurs over the Kuroshio-Oyashio current, south of Greenland, and near the interface between the Atlantic and Arctic oceans (Fig. 1), while elsewhere it does not affect more than 10-15% of the time steps considered."

**4) L 105-106: As the regions of strong surface heat fluxes and high occurrence of CAOs are almost identical, it seems a bit arbitrary/suggestive to associate one with DIAB and the other one with TILT. I'd suggest to change the wording a bit to leave this more open.**

We agree with the Reviewer that regions of strong heat fluxes and most frequent CAOs do substantially overlap. To avoid any premature speculation

at this stage, we have rephrased it as follows: "Both DIAB and TILT are most intense along SST gradients associated with western boundary currents ($7$–$8$ m/km day$^{-1}$ for DIAB, $15$–$17$ m/km day$^{-1}$ for TILT). The same areas of strongest DIAB and TILT also coincide with strong surface heat fluxes (Fig. 1c) and a higher occurrence of CAOs (Fig. 1e)."

**5) L 118: "despite weaker slope": Is it really weaker in the Atlantic?**

The original description was somewhat unclear so we have rephrased it as follows: "The four domains (Fig. 1a, Table 1) represent the upstream (GSE, KOE) and downstream (ENA, ENP) sectors of both the North Atlantic and North Pacific storm tracks. In the upstream regions, the slope features peak intensity in correspondence with strong SST gradients. Over the downstream regions, the slope is more evenly distributed spatially and maxima align with the most intense weather activity (as measured by storm track density, Fig. 1f)."

**6) L 145: I'd suggest to use a different symbol for the velocity in phase space, as u already denotes the wind velocity in equation 2, which confused me in the first place.**

We have changed $\mathbf{u}$ on line 145 into $\mathbf{c}$ to avoid any confusion. We have also rewritten Equation 2,

$$\mathbf{F} = (D\,c_x, D\,c_y) = \Big(-\frac{d\psi}{dy}, \frac{d\psi}{dx}\Big),$$

to include explicitly the phase velocity components $c_x$ and $c_y$, which become useful for a later discussion on how we estimate the duration of a cycle.

**7) L 167: "slightly shorter cycles": This is only true for the western boxes; for the eastern ones, cycles are actually longer in the free troposphere.**

As the difference in duration is not further commented upon, we decided to remove this remark and simply state: "We obtain an average duration for

one revolution of about 5 days, with values ranging between 4.3–5.8d (see Table 2)"

**8) L 172: "increases both with DIAB and TILT": As it is larger for more negative TILT, this statement is technically not correct.**

**L 174: "peaks in the lower-left quadrant": I can see this only for ENP.**

In light of both comments, we have rephrased the text as follows: "The mean isentropic slope in the near-surface troposphere over the GSE region increases both with DIAB and TILT, reaching maximum values around one standard deviation above its time-mean in the lower-right quadrant of the phase space (Fig. 3a), while it increases primarily with TILT and peaks in the lower quadrant in the other regions (Fig. 3b-d)."

**9) L 189-190: I find this sentence quite unclear. If the time reference does not correspond to the typical duration, what does it measure at all?**

The progression along any of the closed trajectories in Figure 3 maps onto time in a non-trivial way. We compute the duration of a cycle by integrating phase speed (i.e., $\sqrt{c_x{}^2 + c_y{}^2}$, where $c_x$,$c_y$ are the horizontal components of the phase space velocity field **c**) along a closed trajectory. However, as the duration of individual events may be shorter or longer, the resulting time duration between different stages is purely indicative and does not necessarily represent the actual time it takes to transition from one stage to the next. Still, the total time duration of a cycle is informative on the time scale associated with the dynamical system, which here is 4–6 days and thus consistent with typical synoptic time scales.

In the revised manuscript, we have rephrased lines 160-161, which now read: "We estimate the average duration for one revolution by integrating the phase speed (i.e., $\sqrt{c_x{}^2 + c_y{}^2}$) along isolines of the streamfunction. For the closed trajectories shown in Fig. 3, we obtain values of about 5d, ranging between 4.3–5.8d (see Table 2)."

We have also rephrased lines 189-190 as follows: "We determined the exact

location of each stage to facilitate the comparison across the four spatial domains. Consequently, time intervals between different stages are somewhat different, ranging between ≈0.7–2d. While the overall cycle duration is indicative of the timescale associated with a dynamical system, the duration of individual events may be shorter or longer. Therefore, the time reference primarily serves the purpose of conveniently tracking evolution along a trajectory, rather than indicating the typical duration of a stage."

**10) L 201: "dominated by TILT": This is not really clear from the figure; the red regions (corresponding to DIAB) are even larger.**

**L 205: "intensifies further upstream": Again, not really obvious to me. There is a lot of overlap and no clear spatial shift.**

**L 208-209: Now I'm totally confused. There are more blue regions (corresponding to TILT) downstream, hence I'd write this sentence the other way around.**

We have rephrased and expanded the first two paragraphs to provide a more accurate description of the composites. In particular, on line 201 we were referring to the increase in TILT that characterises the first stage, as defined from inspection of the phase portraits without considering the composites. On line 205, we were referring to the slight spatial shift between areas of strong TILT and strong DIAB, with the former always slightly to the west of the latter, which is arguably not easy to discern from the figures in the current resolution. We have tried to improve their resolution in the revised manuscript. On line 209, when we say that DIAB gradually spreads downstream, we meant 'downstream' as a direction, that is, DIAB is spreading towards the downstream.

The first paragraph now reads: "In the first stage, which is characterised by the intensification of TILT (see phase portraits, Fig. 3e–h), all of the four domains feature negative anomalies in Z500 and Z1000 (Fig. 5a,b and Fig. 6a,b), indicating advection of cold air from continents for KOE and GSE and from polar oceans over warmer ocean surfaces for ENP and ENA. The structure of the flow is consistent with the onset of CAOs, which are most frequent in these regions in winter (Grønås and Kvamstø, 1995; Dorman et al., 2004; Kolstad et al., 2009). The spatial distribution of strong TILT trails behind the advancing cold air front, while DIAB intensifies upstream of peaks in TILT (i.e., to their west). Strong DIAB and TILT outside of the averaging domain, such as over the Davis Strait for GSE composites or

south of the Bering Strait for KOE composites (Fig. 5), likely reflect their climatological mean rather than a specific relevance to a particular stage in the phase portrait."

The first half of the second paragraph now reads: "According to phase portraits, the second stage is characterised by the steepest slope, as TILT reaches its maximum while DIAB is still increasing. Composites for this stage (Fig. 5c,d and Fig. 6c,d) show a strengthening and, especially in the ENA region (Fig. 6c), a downstream progression of the cyclonic circulation that emerged in the previous stage. We observe again that the spatial distribution of TILT features a shift to the east with respect to that of DIAB, which is now dominant and gradually spreading westwards."

**11) L 213-214: TILT also prevails in the regions of the boundary currents.**

While it is true that TILT retains some of its intensity, it is still weaker than that observed in the previous stage (Fig. 5c,d) and distributed over a smaller area compared to DIAB. We have rephrased those two lines avoiding the term 'prevail', which was perhaps confusing. We have also specified that the ENP region behaves differently. "TILT has subsided compared to the previous stage, while DIAB has retained its strength. The picture is somewhat different for the ENP region, where TILT does not appear to have changed much while DIAB is visibly stronger."

**12) L 215: "anticyclonic geopotential anomalies start building up": not so much in the Pacific, at least at this stage**

We have added the following: "except over the KOE region, where positive anomalies have not yet formed at this stage."

**13) Section 5.1: The more general conclusions from this section are not clear to me. I got a bit lost in the details, which, in addition, are not always consistent between text and figures (as noted above).**

We have added a summarising paragraph that hopefully clarifies the main conclusions that can be drawn for the near-surface. A summary of both near-surface and free troposphere is provided in the concluding section as well.

"In the near-surface, we can therefore ascribe the particular phasing between DIAB and TILT to the effect of cold air advection associated with CAOs as well as cold sectors of midlatitude weather systems. The propagation of the cold front associated with these events contributes to a local steepening of the slope. Steep slope prompts an almost instantaneous response in TILT, whereby isentropic surfaces are flattened as cold air masses sweep in over the ocean surface. The thermal contrast between the cold air masses and the warm ocean surface eventually triggers surface heat fluxes that force the isentropic surfaces back to their initial position, thus diabatically restoring the near-surface slope."

14) **L 225: "particularly in correspondence with the upper-level anomalies": Again not that obvious; e.g., in 7a DIAB is clearly shifted towards the lower-level anomaly.**

We have rephrased that sentence to provide a more accurate description. We meant to highlight the spatial link between peaks in DIAB and the upper-level anomalies to which they seem more closely connected. DIAB shown here is integrated across the free troposphere, not the near-surface.

"DIAB is most active in close proximity to these cyclonic anomalies, particularly on the southern and western flanks of the upper-level anomalies, with the exception of the ENA region (Fig. 7a), where DIAB is somewhat weaker and peaks to the north of the upper-level anomaly. TILT, on the other hand, remains negligible across the four regions."

15) **Section 5.2: Again, the general conclusions could come out more clearly. Most of the corresponding statements are quite generic (diabatic processes are important for midlatitude waves, L 246; primary importance of latent heat release in the diabatic restoration, L 256; role of moist diabatic processes in the evolution of cyclones, L 263) and already quite well known from previous studies (also from your own group...). What are really the new aspects**

**from this analysis?**

Also in response to the first Reviewer, we rephrased a few paragraphs to make our conclusions more evident and to highlight the new aspects that we show in this study, namely the particular phasing between the diabatic and tilting tendencies and the physical mechanisms behind it. Our results are consistent with previous work by Papritz and Spengler (2015) and Weijenborg and Spengler (2020), adding to a growing body of literature on the crucial role of moist diabatic processes in the representation of storm tracks (Willison et al., 2023; Schemm, 2023; Fuchs et al., 2023).

16) **L 277: "DIAB takes place ahead of storm activity, both in time and space": I'm not sure that this conclusion is justified. For instance, over the North Atlantic (Figs. 7a,c; 8a,c), both cyclonic anomalies and DIAB develop in parallel, and spatially quite well aligned.**

We were primarily referring to the upper-level anomalies (Z500), where DIAB is more markedly ahead/downstream. It was somewhat misleading so we have rephrased it as follows: "Composites across the phase space confirm that DIAB is tightly linked to the development and further evolution of storm activity, both in time and space (Figs. 7,8)"

17) **Figure 5: Should the caption read "below -15... for TILT"? Also, 15 seems to be quite high for DIAB when compared to Fig. 3; is this really correct? Indicate in the caption what the numbers on the sides of the plots indicate (mean DIAB/TILT).**

A previous version of the manuscript had "above 15... in magnitude" which would have implied below -15 for TILT. We have changed the caption accordingly and also included the meaning of DIAB and TILT to the left and right, respectively, of each panel.

The values for DIAB might seem quite high, especially with respect to climatology. However, these peaks occur over a limited spatial extent compared to the entire domain over which the spatial average is computed.

---

## Author Response (AR2)

**Diabatic effects on the evolution of storm tracks**

Marcheggiani and Spengler

**Authors response**

**Thanks for the detailed responses and helpful revisions of the manuscript. In my opinion, the manuscript is now ready for publication. I only have a few technical comments, as detailed below.**

We are happy to hear that we have addressed the concerns raised by the Reviewer and would like to thank them again for reviewing our work.

**Just one additional general remark: One thing that has become clearer to me through the revision is that the advection term does play an active role in your storyline, in particular, in the lower troposphere, as, for instance, discussed in the responses to the other reviewer and the revised conclusions ("cold air masses bring anomalously steep slope into the spatial domain"). Maybe, in your future research, you may consider this term more explicitly in the budget; but I don't think this is required for the present manuscript.**

Yes, we agree that , at the surface, the advection term becomes more relevant in understanding the dynamics behind the phase space circulation. We plan to look into cold-air outbreaks and how the slope diagnostics evolve during these events, which we believe are the main contributors to the average picture we have gathered so far.

**Technical comments:**
**(line numbers refer to the manuscript version with tracked changes)**

**L 194: I still think that this formulation is unclear. "increases both with DIAB and TILT" would mean that the highest slope values are obtained for high DIAB and high TILT, while actually they are found for low (strongly negative) TILT. Please rephrase.**

TILT is mostly negative, so we refer implicitly to its magnitude when we say it increases. We understand it might result confusing at times so we have rephrased relevant lines to make it more explicit: "The mean isentropic slope in the near-surface troposphere over the GSE region increases with the magnitude of both DIAB and TILT, …"

**Also more in general, the wording is not always clear in this context, for instance, if you refer to a "maximum" in TILT and actually mean the lowest values (technically a minimum), or "TILT has subsided" when it approaches zero, but actually increases (due to the dominantly negative values).**

We primarily refer to the magnitudes of these terms and in that sense the verb 'subside' is perhaps the most appropriate to use, as we focus on intensity. Therefore, we have decided to keep 'subside'. However, we have rephrased the text where we refer to an increase in TILT to say explicitly that it is its magnitude/intensity we are referring to.

**L 196: "lower quadrant" is also a bit awkward, because there are two lower quadrants.**

We changed 'lower quadrant' to 'lower half'.